# Predicting performance in 4 x 200-m freestyle swimming relay events

**Paul Pao-Yen Wu**[1,2]*, **Toktam Babaei**[1,2], **Michael O'Shea**[1,2], **Kerrie Mengersen**[1,2], **Christopher Drovandi**[1,2], **Katie E. McGibbon**[3], **David B. Pyne**[4], **Lachlan J. G. Mitchell**[3], **Mark A. Osborne**[5]

**1** School of Mathematical Sciences, Queensland University of Technology, Brisbane, QLD, Australia, **2** ARC Centre of Excellence in Mathematical and Statistical Frontiers (ACEMS), Melbourne, VIC, Australia, **3** Queensland Academy of Sport, Nathan, QLD, Australia, **4** University of Canberra Research Institute for Sport and Exercise, Bruce, ACT, Australia, **5** School of Human Movement & Nutrition Sciences, University of Queensland, Brisbane, QLD, Australia

* p.wu@qut.edu.au

## Abstract

### Aim

The aim was to predict and understand variations in swimmer performance between individual and relay events, and develop a predictive model for the 4x200-m swimming freestyle relay event to help inform team selection and strategy.

### Data and methods

Race data for 716 relay finals (4 x 200-m freestyle) from 14 international competitions between 2010–2018 were analysed. Individual 200-m freestyle season best time for the same year was located for each swimmer. Linear regression and machine learning was applied to 4 x 200-m swimming freestyle relay events.

### Results

Compared to the individual event, the lowest ranked swimmer in the team (-0.62 s, $CI = [−0.94, −0.30]$) and American swimmers ($−0.48$ s $[−0.89, −0.08]$) typically swam faster 200-m times in relay events. Random forest models predicted gold, silver, bronze and non-medal with 100%, up to 41%, up to 63%, and 93% sensitivity, respectively.

### Discussion

Team finishing position was strongly associated with the differential time to the fastest team (mean decrease in Gini (MDG) when this variable was omitted = 31.3), world rankings of team members (average ranking MDG of 18.9), and the order of swimmers (MDG = 6.9). Differential times are based on the sum of individual swimmer's season's best times, and along with world rankings, reflect team strength. In contrast, the order of swimmers reflects strategy. This type of analysis could assist coaches and support staff in selecting swimmers and team orders for relay events to enhance the likelihood of success.

**Data Availability Statement:** All relevant data are included within the manuscript and its supporting information files.

**Funding:** This research was conducted by the Australian Research Council Centre of Excellence for Mathematical and Statistical Frontiers (project number CE140100049) and funded in part by the Australian Government. It was also supported by the Queensland Academy of Sport's Sport Performance Innovation and Knowledge Excellence unit, and by Swimming Australia Limited. Funding was awarded for the project, not to authors Grant numbers - NA URLs: https://acems.org.au/home https://www.qld.gov.au/recreation/sports/academy/services/spike https://www.swimming.org.au/ NO The funders had no role in study design, data collection and analysis, decision to publish, or preparation of the manuscript.

**Competing interests:** The authors have declared that no competing interests exist.

## Introduction

A key challenge of relay events in sporting competitions is team selection and the order of athletes, as they can impact race outcomes [1–3]. In the 4 x 100-m track running relay, factors such as the preferred hand for giving and receiving a baton, athlete skills in running bends, and the lane drawn, should all be considered when selecting team order [1]. In swimming, key considerations for relay team performance include flying start technique and exchange block times [4–6]. As a result, much of the relay literature in swimming focuses on technical components related to the dive start, including differences between the flat start performed in individual events and the flying start performed by swimmers assigned to the second to fourth relay positions [7, 8]. However, in both track running and swimming, it appears that selecting the fastest athlete for the first or lead-off relay leg is popular and successful [1–3], although further research is required to determine how this impacts team performance.

Recently, pacing differences between individual and relay events in swimming have been examined indicating that some swimmers alter their pacing strategy during relay events [9]. This difference in pacing strategy between individual and relay swims may be attributed to the relay leg assignment as well as the added pressure to perform well for the team [9]. Extensive research on team dynamics and behavioural aspects of competitive relay swimming are described in the literature [10–13]. Compared to individual events, swimming performance is typically faster in relays which may be attributed to elevated motivation and effort [13, 14]. However, there is conflicting evidence of differences in starts, turns and swimming speed between individual and relay events [8]. In addition to the motivational effects of relay swimming, the order of swimmers in the relay team can also potentially impact the effort exerted by each swimmer. Swimmers positioned in later relay leg positions were found to be more likely in certain contexts to swim faster than those in earlier positions relative to individual event times [11, 12]. Contextually, the positive influence of relay leg positioning has been ascribed to an increase in the perceived importance of individual contributions to the team outcome [12, 14].

During FINA-sanctioned events including the biennial World Championships, relay teams must nominate their four selected swimmers, and the team order, one hour prior to the start of the heats or finals session in which the relay occurs [15]. However, swimmers are typically selected for the national squad a few weeks to several months prior to the competition based on their performance in the corresponding individual event. In addition, due to the complex interactions between physiological, psychological and team-based dynamics [10], predictions of individual performance in relays and overall team outcomes are challenging. Therefore, there is a need for effective predictive tools that could support coaches in the decision-making process to maximise the performance of the relay team as a whole, as well as each individual swimmer.

With an abundance of performance data now available in many sports as a result of advances in technology, data-driven models are becoming increasingly popular in sports science [16–19]. These models have been applied to swimming in an attempt to predict the performances of individual events based on training and competition data, as well as anthropometric, physiological and biomechanical characteristics. Despite extensive analyses of behavioural and other components of relay swimming, limited work to date have brought together the various components to predict race outcomes of relay events. The 4 x 200-m freestyle relay is currently the longest relay in the FINA competition schedule. With each swimmer required to complete 4 x 50 m laps the event requires well-developed speed-endurance, technical skills in starts, turns and finishes, and the element of pacing and sufficient data to model pacing effects [9]. This complexity makes the 4 x 200-m freestyle ideal as a starting point for

developing and testing predictive and analytical tools. Therefore, the aim of this study was to predict and better understand contextual factors contributing to relay team performance in light of individual swimmer performance, and develop predictive models to analyse the relationship between a team's finishing position and these factors for the 4x200-m swimming freestyle relay.

## Methods

### Data collection

Race data for 4 x 200-m freestyle relay finals from 14 international long course competitions between 2010 and 2018 were analysed retrospectively, comprising Olympic Games (2012, 2016), Pan Pacific Championships (2010, 2014), World Championships (2011, 2013, 2015, 2017), Commonwealth Games (2014, 2018) and European Championships (2010, 2012, 2014, 2016). This data set has been used previously [9] and included the world ranking for each swimmer in the year of that competition, reaction times, 50-m splits and overall times. For each relay swimmer, the individual 200-m freestyle season's best time for the same season (typically beginning around September and concluding around July-August) was located using FINA world rankings (https://www.fina.org/content/swimming-world-ranking). A total of 716 relay swims, divided approximately evenly across the four relay legs, and corresponding individual event swims were analysed across 348 different swimmers (175 males and 173 females). To allow comparisons between individual and relay events, exchange block times for relay swimmers positioned on the second to fourth leg were adjusted to equal reaction times from the individual event using the methodology developed by [9] and [20].

The team average ranking was calculated as the average world ranking of the four swimmers in the team, where world rankings for the year of the relay competition were used. In our dataset, the swims contributing to the rankings were coincidentally prior to the major competition of that year. The best or highest ranking and the worst or lowest ranking within the team was used as an indicator of team depth. Relay team data was only considered when individual season's best times for all four swimmers within the team were available, resulting in the analysis of 121 teams of a total of 188. Relay teams were classified into four categories by finishing position: (1) gold, (2) silver, (3) bronze, and (4) a non-medal position (4th to 8th placed teams). There were 20, 17, 16, and 68 data points for each of these finishing positions, respectively. Team characteristics (i.e. explanatory variables) included world ranking of individual swimmers and team average ranking, the order of swimmers in the team, individual season's best 200-m freestyle time, start lap strategy, and pacing strategy from the relay race (Table 1). To capture the potential differences in preparation, performance level and the number of nations competing at the World Championships and Olympic Games compared to other events such as Commonwealth Games, a variable to capture competition effects was included in the model. Furthermore, to better understand trends across nations while preserving model identifiability [21], we included a categorical variable to specify two major competitor nations, namely the USA and Australia, whereas other nations were grouped into a third category of 'Other'.

The order of swimmers in the relay was encoded according to the relative world ranking of each swimmer within a team. For example, a relay order of "2-1-3-4" indicates that the second fastest swimmer swam (i.e. second highest world ranking) the lead-off or first leg, the fastest swimmer swam the second leg, and so on. However, the large number of swimmer permutations, including order as a categorical variable, directly leads to problems with model identifiability. As a result, the order categorisation only considered the unique positions for the first and second fastest swimmers. In this scheme, a swimming order of "2-1-3-4" and "2-1-4-3"

**Table 1. Description of the explanatory variables.**

| Variable | Description |
|---|---|
| Gender | Female or male |
| Nation | Nationality of the team/swimmers. There were a total of 33 nations categorised into three groups: USA, Australia and Other |
| Season's Best time | Fastest individual 200-m freestyle time within the same season |
| Relay time | Time taken for an individual swimmer to complete their 200-m freestyle relay leg |
| Team performance time | Sum of the relay times for the four swimmers on each relay team. Total time taken to complete the race by the team |
| World Ranking | Ranking according to FINA based on the individual season's best time for the 200-m freestyle |
| Relay Leg | The position of the swimmer within the relay team e.g. 1, 2, 3 or 4 |
| Order of the team | Chronology of swimmers according to relative world ranking within the team |
| Team average ranking | The average of the world ranking of the four swimmers in the team |
| Best Ranking | The highest world ranking in the team (fastest swimmer) |
| Worst Ranking | The lowest world ranking in the team (slowest swimmer) |
| Finishing Position | Finishing place of the team in a relay final (1–8). Classified into four categories: |
| | (1) Gold |
| | (2) Silver |
| | (3) Bronze |
| | (4) Non-medal position (4th-8th position) |
| Start lap strategy | Percentage of race time spent in lap 1 categorised as average, fast or slow |
| Pacing strategy | Slope of laps 2–4 and laps 3–4 categorised as even, negative or positive |

are both represented as "21xx". In view of the many permutations of ordering that have not been widely used, we focused on the five most frequently employed permutations ("1x2x", "1xx2", "21xx", "2xx1", "12xx") and included a category 'other' for the remaining permutations. Pacing was determined from the swimmer's best individual performance that season by converting split times into a percentage of overall race time spent in each 50-m lap, and characterised by the start lap strategy (lap 1) and pacing strategy (laps 2–4) [9].

## Statistical analysis

Two main types of methods were used: (i) linear regression, to study individual swimmer's relay performances, and (ii) random forests, to predict race outcomes given team configurations. This section describes the two methods, model fitting and model validation.

**Linear regression.** Multiple linear regression [21] was used to estimate the relationships between an individual swimmer's performance in a relay and the explanatory variables (Eq 2). An explanatory variable was deemed to have a significant effect if $p \leq 0.05$.

**Random forests.** Random forests were used to predict team finishing positions based on explanatory variables as they are ideally suited for a mixture of numeric and categorical variables with potentially highly non-linear relationships. Random forests are an ensemble modelling extension of simple decision trees, which recursively partition the space of explanatory variables to minimise some dispersion criteria (i.e. measure of variability) in the resultant partitions [22]. Random forests have also demonstrated high predictive sensitivity and specificity for complex problems in many domains [22]. This method helps to overcome the overfitting problem encountered in decision trees by building many shallow trees using data subsets sampled through bagging. We built a random forest model, referred to as RF1, to predict gold, silver, bronze or non-medal finishing positions. To assist with better prediction of medal colour, we also trialled a

model that only predicts medal colour, RF2. We developed a predictor variable based on the observation that team performance in a relay is the sum of the individual performance times of the four swimmers within the team. Therefore, based on the sum of the season's best individual times we constructed a theoretical performance measure of each team relative to the theoretical performance of the fastest team based on differential time (*Diff.Time*) defined as follows:

$$Diff.Time_j = \sum_{i=1}^{4} s_{ij} - \min_{\forall j} \sum_{i=1}^{4} s_{ij} \tag{1}$$

where for team *j* and individual *i*, $s_{ij}$ is the season's best time for that swimmer.

**Model fitting.** All statistics were calculated using R software [23] and implemented with the base and randomForest packages to fit linear regression and random forest models, respectively. The parameters of the random forest were tuned by making use of a cross-validation based technique. Five-fold cross validation was run 100 times in conjunction with a grid search for selecting model parameters including the number of variables to sample at each split in the tree, and the number of variables sampled as candidates at each split in the tree. Given the randomly sampled nature of random forests, repeated evaluations provide a more robust selection for the tuning parameters [24].

**Model validation.** For the linear regression model, goodness of fit is sufficient to give confidence that the model is reasonable, and the model can be interrogated to ascertain the impact of different explanatory variables on individual performance [21]. In comparison, the random forest was employed to predict race finishing position, so we validated model performance using leave-one-out cross-validation. In this scheme, we iterated over each data point, trained with all other data points and tested with the current data point.

We used a 4x4 confusion matrix to show the number of times a recorded gold, silver, bronze or non-medal result (corresponding to rows) was classified by the model as a gold, silver, bronze or non-medal outcome (columns corresponded to predictions). We computed model sensitivity, also referred to as producer's accuracy when there are more than two categories, which is the rate at which the model correctly classifies a result as a member of a certain category [24]. Note that there is no direct analogue for specificity when there are more than two categories. The randomForest package uses the Gini index as one approach to capture both sensitivity and specificity [22]. This index is useful for assessing both the validity of the model, and for quantifying the relative influence of explanatory variables based on the decrease in the Gini index when a variable is removed from the model.

Finally, the utility of the random forest was demonstrated by applying it to a case study analysis of the 2019 World Championships.

## Results

### Variables affecting swimmer performance in the relay

The multiple linear regression model had a $R^2$ goodness of fit value of 0.97 which explains 97% of the variation in the data. The model formulation was:

$$Relay.Time_i = Nation + Gender + Season's.Best_i + Relay.Leg_i + Team.Rank_i + Pacing.Start_i + Pacing_i + \epsilon \tag{2}$$

where swimmer *i*'s relay swim time is explained by nation, gender, season's best individual 200-m freestyle event time, relay leg assignment, relative world ranking within the team (1 for highest, 4 for lowest), start lap strategy and pacing strategy corresponding to the individual's season's best performance, and $\epsilon$ is a normally distributed error term. The predictor variables

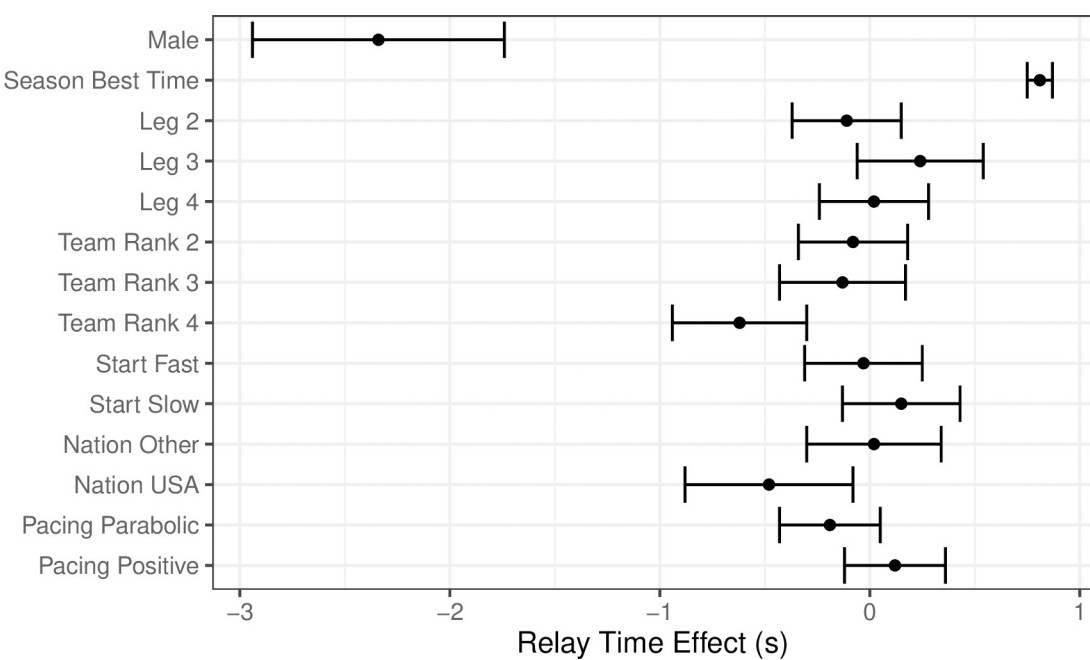

**Fig 1. Coefficients for explanatory variables and individual swim performance in the relay.** Note that the baseline categorical variable for gender is female, for relay leg is the first leg, team ranking is rank one, pacing start lap strategy is average, nation is Australia, and pacing strategy is even. *If the confidence interval does not overlap with 0, then the effect is considered significant.*

explained 97% of the variation in relay time, and were selected to investigate team selection (e.g. swimming rankings), relay leg and ordering, and pacing effects as discussed in Section 1. Noting that all covariates are categorical with the exception of one, assumptions of normality and independence were checked with a residual plot (S1 Fig).

The resulting model coefficients are presented in terms of their mean effect and 95% confidence interval (CI), on a scale of seconds (Fig 1). The model coefficients show that males are on average 2.34 s faster than females per leg in the 200m relay event. Thus, having accounted for other variables in the model such as gender and nation, the fourth ranked swimmer (slowest) appears to perform faster than expected by -0.62 s (95% confidence interval CI = [-0.94,-0.30], p = 0.00) compared to the first ranked swimmer. In addition, compared to the first leg, swimmers in the third leg tend to swim slower than expected by 0.24 s (CI = [-0.05,0.54], p = 0.10). While start lap strategy in the individual 200-m freestyle event did not appear to impact relay time, swimmers who displayed a parabolic pacing strategy in their individual event can potentially swim faster than those with an even (-0.19 s, CI = [-0.43,0.05], p = 0.11) and positive (0.12 s, CI = [-0.12,0.35], p = 0.31) pacing strategy; these effects were again not significant, but potentially of interest for future investigation. Swimmers from the United States typically swam -0.48 s (CI = [-0.89,-0.08], p = 0.02) faster than expected compared to swimmers from all other nations. However, there do not appear to be any significant effects attributed to the relay leg.

## Predicting team finishing position

Through parameter tuning, a random forest model that sampled two variables at each split was selected with the following formulation:

$$Estimated.Finishing.Position = f \left( \begin{array}{l} Gender, Nation, Order, Team.Avg.Rank, Best.Rank, \\ Worst.Rank, Diff.Time, Elite.Competition \end{array} \right) \quad (3)$$

**Table 2. RF1 misclassification matrix of medalling teams: Gold, Silver, Bronze; and non-medalling teams.**

| | | Model Predicted Output | | | | | |
|---|---|---|---|---|---|---|---|
| | | Gold | Silver | Bronze | Non-Medal | Total | Sensitivity |
| True Finishing Position | Gold | 20 | 0 | 0 | 0 | 20 | 1.00 |
| | Silver | 3 | 6 | 2 | 6 | 17 | 0.35 |
| | Bronze | 1 | 2 | 2 | 11 | 16 | 0.13 |
| | Non-Medal | 0 | 1 | 4 | 63 | 68 | 0.93 |
| | Total | 24 | 9 | 8 | 80 | | |
| | User's Accuracy | 0.83 | 0.67 | 0.25 | 0.79 | | |

where a team's finishing position is predicted using gender, nation, order of swimmers, team average ranking, best and worst world ranking, *Diff.Time* (differential time) and race type (elite competition).

Cross-validation results for RF1 show that gold medal predictions achieved 100% sensitivity (producer's accuracy) and non-medalling results were predicted with 93% sensitivity (Table 2). The most influential variables in the model and their impact on cross-validated predictions, measured using Mean Decrease Gini (MDG), are: differential time (MDG = 31.3), team average ranking (18.9), best ranking (12.6), order of swimmers (6.9), nation (2.6) and gender (1.8) [22]. However, it was significantly harder to predict silver and bronze race outcomes as the model achieved sensitivities of 35% and 13%, respectively. In contrast, cross-validation of RF2 predictions of medal colour achieved sensitivities of 100%, 41% and 63% for gold, silver and bronze finishing positions, respectively (Table 3).

The practical validity of the prediction model was tested by predicting the finishing positions of the top four male and female teams at the 2019 FINA World Championships. The probability of each team achieving a gold, silver or bronze medal was predicted in model 1, and the probability of finishing in a medal or non-medal position in model 2 (Table 4). We also modelled the probability of various team orders showcasing the battle for the bronze medal position in the Women's 4 x 200-m freestyle relay (Table 5). Canada was successful in winning bronze despite having a lower probability of success across the various team orders.

## Discussion

The statistical approaches developed in this study were useful in identifying the variables affecting relay swimming performance given individual swimmer performance, and predicting relay team finishing positions for the 4x200-m freestyle relay. Results indicate that swimmers from the USA, and those swimmers who were the slowest within their teams according to ranking, typically performed better in relays than in individual events. The random forest

**Table 3. RF2 misclassification matrix of medalling teams into three groups of Gold, Silver and Bronze.**

| | | Model Predicted Output | | | | |
|---|---|---|---|---|---|---|
| | | Gold | Silver | Bronze | Total | Sensitivity |
| True Finishing Position | Gold | 20 | 0 | 0 | 20 | 1.00 |
| | Silver | 3 | 7 | 7 | 17 | 0.41 |
| | Bronze | 1 | 5 | 10 | 16 | 0.63 |
| | Total | | 24 | 12 | 17 | |
| | User's Accuracy | | 0.83 | 0.58 | 0.59 | |

**Table 4. Probability output of the random forest model for the top four female and male 4 x 200-m freestyle relay teams in the 2019 FINA World Championships.**

| Finish Position | Nation | Nation Category | Time (min:sec) | Gender | Ranking | | | Order | RF1 (Pr) | | | |
|---|---|---|---|---|---|---|---|---|---|---|---|---|
| | | | | | Team Avg | Best | Worst | | Gold | Silver | Bronze | Non- Medalling |
| Gold | Australia | Australia | 7:41.50 | F | 9 | 2 | 17 | 1xx2 | 0.84 (+) | 0.10 | 0.06 | 0.00 |
| Silver | USA | USA | 7:41.87 | F | 18 | 7 | 43 | 21xx | 0.08 | 0.44 (+) | 0.35 | 0.13 |
| Bronze | Canada | Other | 7:44.35 | F | 24 | 12 | 41 | Other | 0.02 | 0.17 | 0.55 (-) | 0.27 |
| 4th | China | Other | 7:46.22 | F | 24 | 6 | 36 | 1x2x | 0.02 | 0.14 | 0.66 (-) | 0.19 |
| Gold | Australia | Australia | 7:00.85 | M | 17 | 2 | 36 | 12xx | 0.71 (+) | 0.13 | 0.17 | 0.00 |
| Silver | Russia | Other | 7:01.81 | M | 14 | 6 | 23 | 2xx1 | 0.33 | 0.40 (-) | 0.25 (-) | 0.02 |
| Bronze | USA | USA | 7:01.98 | M | 19 | 11 | 27 | 1xx2 | 0.23 | 0.51 (-) | 0.20 (-) | 0.07 |
| 4th | Italy | Other | 7:02.01 | M | 39 | 10 | 70 | 12xx | 0.01 | 0.02 | 0.15 | 0.81 (+) |

F = female, M = male, Pr = probability.

Note: Correct predictions are annotated with (+) and incorrect predictions with (-).

model RF1 was highly effective at correctly predicting gold medal winning teams (100% sensitivity), and whether a team will medal or not (non-medalling sensitivity of 93%). However, the models were less accurate in distinguishing between silver (35% using RF1, 41% using RF2) and bronze (13% using RF1, 63% using RF2). This outcome might be due to small differential times between these positions for some swimming competitions. In contrast, the differential times between the bronze medal position and non-medal positions for all competitions tended to be much larger. The RF2 model could be used by decision makers to evaluate silver and bronze medal scenarios assuming that a team will win a medal. These models enable coaches and support staff to simulate different relay race scenarios to determine the optimal relay team configuration by using swimmer characteristics, anticipated opponent swimmers and team order.

**Table 5. Effect of team order on the probability (Pr) of Gold, Silver, Bronze or no medal for Canada and China who finished 3rd and 4th, respectively, in the women's 4 x 200-m freestyle relay at the 2019 FINA World Championships.**

| Team | Pr (Gold) | Pr (Silver) | Pr (Bronze) | Pr (NoMedal) |
|---|---|---|---|---|
| Bronze (3rd Place)—Canada | | | | |
| 1xx2 | 0.004 | 0.247 | 0.385 | 0.364 |
| 21xx | 0.002 | 0.164 | 0.447 | 0.387 |
| other | 0.015 | 0.173 | **0.547** | 0.265 |
| 2xx1 | 0.009 | 0.267 | 0.334 | 0.390 |
| 12xx | 0.002 | 0.216 | 0.375 | 0.407 |
| 1x2x | 0.006 | 0.188 | 0.523 | 0.283 |
| 4th Place—China | | | | |
| 1xx2 | 0.001 | 0.219 | 0.542 | 0.229 |
| 21xx | 0.005 | 0.098 | **0.712** | 0.185 |
| other | 0.021 | 0.135 | 0.667 | 0.177 |
| 2xx1 | 0.012 | 0.250 | 0.453 | 0.285 |
| 12xx | 0.006 | 0.147 | 0.621 | 0.226 |
| 1x2x | 0.017 | 0.137 | 0.659 | 0.187 |

Note: Pr values in bold text indicate the team order with the highest probability of winning the bronze medal and rows shaded in grey indicate the team order used in the race

## Differentiating psychological from technical effects

Among the many variables that may impact relay swimming performance, the psychology of team competition is important [13, 14]. Note that we have adjusted for the effect of the flying start in relay legs two through four by setting exchange block times equal to individual reaction time [8]. Any residual differences between legs were captured via the relay leg term; thus, we were able to discern potential psychological effects from technical effects.

Our results indicate that the largest effect of the variables modelled in this study was due to the worst-ranked or slowest swimmer in a team. These swimmers typically swam 0.62 s faster in the relay than in the corresponding individual event. Peer effects can have a positive impact on individual performance within a team, and these psychosocial effects may help explain the improved performance of some swimmers in relays relative to their individual times in the present study [10, 25]. However, our findings differ from those of Hüffmeier and Hertel [12] who reported on the effects of relay leg assignment (i.e. going first, second, third or last). In contrast, we found the relative ranking of the swimmer within the team (i.e. worst-ranked swimmer) to be a larger effect, and relay leg assignment to be generally not significant. Motivating group effects are typically greater when an individual perceives their contribution as important to the overall team outcome [12, 14]. Therefore, it is possible that the slower swimmers within the team felt more pressure and motivation to step up and put their team in a good position. In contrast, relay teams comprised of higher ranking athletes are more likely to underperform relative to their individual performance [25]. Such psychological impacts could be an area for further study to help motivate and develop swimmers in relay and non-relay contexts.

## Nationality impacts

Swimmer nationality also impacted performance as individual swimmers from the USA tended to swim 0.48 s faster during the relays than their predicted individual swim times. This outcome could be attributed to the competition structure of the National Collegiate Athletic Association (NCAA) which allows for the frequent practise of relay swimming in competitive races. In contrast, Australian swimmers (and those of many other nations) may only swim in a limited number of relay events throughout the season prior to the major international competition, and rarely get the opportunity to practice with potential teammates. Team cohesiveness may play a role as social loafing is less likely to occur in highly cohesive teams [26]. However, further research is required to determine the underlying nature of differences between nations.

## Relative influence of variables

The ability to accurately predict team finishing position based on a set of explanatory variables would support coaches in making an evidence-based decision when selecting relay team swimmers and leg assignments, potentially weeks to months ahead of competition. Random forest models were used to make these predictions and the most influential variables were identified based on cross-validation, and the mean decrease in sensitivity and specificity as measured by MDG [22]. As might be expected, the strength of the team, as captured by rankings and individual season's best times, was the leading contributor to finishing position (Results). However, team strategy, in terms of the order of swimmers was the next most influential factor. The dataset used for modelling comprised primarily of high calibre, international events including Olympics and World Championships. Typically, these are the pinnacle events that athletes train and prepare for. We identified that medal outcomes were highly influenced by differential time (MDG of 31.3), which is based on the sum of individual swimmer's season's

best times. This outcome suggests that individuals are performing at or near their best at these international relay competitions and, equivalently, that season's best times are useful in predicting individual swimmers' performance at pinnacle relay events.

## Illustrative case study

To illustrate how the model could be used to support decision making, we demonstrate with a case study of predicting the finishing positions for the top four teams at the 2019 FINA World Championships. This data, which included world rankings and season best times coming into the competition, were not included in the original dataset. Although the gold medal predictions were correct, the model incorrectly predicted the bronze and 4th positions for females, and silver and bronze positions for males. Team average ranking for the two female teams was identical with a similar differential between the highest and lowest ranked swimmer. However, the fourth placed team had the best ranking swimmer overall, which may indicate that this team underperformed relative to their expected team performance time. This explanation may also serve as a reason for the incorrect model prediction here for both medal colour and medal or non-medal. Similarly, the USA Men's team was predicted to finish in the silver medal position, but Russia outperformed them by just 0.17 s. However, the model was able to correctly predict a medal and non-medal position.

These models can also be used in a predictive decision support scenario where the impact of different swimmer orders on finishing position can be evaluated in a risk-informed, probabilistic manner. For the Canadian women's teams in the 2019 FINA World Championships, the order used in the race provided the highest chance of bronze and lowest chance of a non-medal finish. A 2xx1 order could have increased the chance for a silver medal by 9.4%, but also increase the chance for missing out on a medal by 12.5%. According to the model, China would have increased their chance of a bronze medal and slightly decreased their chance of a non-medal finish if they applied another swimmer order or 21xx. However, these scenarios only serve as illustrations, and should be seen as observations given limitations of the data, the numerous possible swimmer order and ranking combinations, and the many other factors influencing medal finishes that were not included in the model.

## Limitations and future work

While these statistical approaches were successfully applied to enhance our understanding of the variables impacting both individual and team performance in relay swimming events, there are some limitations. First, only teams with available data for all four swimmers were analysed, resulting in partial data for some races which is a potential source of misclassification errors. However, increasingly more data are becoming available as demonstrated by the availability of each swimmer's season's best time and world ranking going into the 2019 World Championships. Data about individual swimmer's physical status or performance characteristics (such as individual and relay block times [20]) could be used to extend this work and improve the predictivity of the model. Currently, less than 25% of the swimmers in this dataset had 3 or more relay races recorded and the majority only had 1; this limited our ability to model individual characteristics.

We also assumed that all relay teams had an equal chance of finishing in each position, whereas in reality some teams are more likely to be chasing medal positions than others. Incorporating such prior knowledge into the model, such as through a Bayesian formulation, could strengthen the prediction accuracy. Moreover, the individual analyses could be combined through a hierarchical model to enable 'borrowing of strength' between events to improve estimates and extend insights. Additionally, intermediary positions and whether the team is still

in medal contention or not at the end of each leg could also affect individual swimmer performance. Multivariate extensions of the proposed models could be developed, with sufficient data, for this more complex task. Finally, hybrids of machine learning and statistical methods could be used in future research to build similar predictive models for other swimming relay events, such as the mixed relays [3].

## Conclusion

The prediction models of this study indicate that the slowest swimmers within a team, and swimmers from the USA, tend to swim faster than expected in relay events. These swimmers can step-up and perform above expectations in relay events relative to their season's best individual event performance. Gold medal and non-medal finishing positions can be accurately predicted by using random forest models, however these models are less accurate in differentiating between silver and bronze medal positions.

The outputs of machine learning models developed in this study can be used by coaches and support staff to assist with decision-making processes around team selection, and for determining the best combination of swimmers to maximise team performance. Different team configurations can be inserted into the model to examine how the probability of finishing in each position changes with different swimmers and team orders. The models can be integrated into decision-making algorithms and expert systems which can also be updated with new data. These prediction models could also be applied to other sports such as track running and cycling where athlete selection and order likely influence team performance.

## Supporting information

**S1 Fig. Residual and normality plot for the linear regression model reported in results.** Note that the residuals are mostly normal and randomly distributed. Although there is a little deviation from normality in the right tail of the distribution (ignoring the outlier), there are relatively few data points here and these slow swim times are not as relevant in the context of predicting medalling performances.
(DOCX)

**S1 File.**
(CSV)

## Author Contributions

**Conceptualization:** Paul Pao-Yen Wu, Kerrie Mengersen, Christopher Drovandi, Katie E. McGibbon, David B. Pyne, Lachlan J. G. Mitchell, Mark A. Osborne.

**Data curation:** Paul Pao-Yen Wu, Toktam Babaei, Katie E. McGibbon.

**Formal analysis:** Paul Pao-Yen Wu, Toktam Babaei, Michael O'Shea, David B. Pyne, Lachlan J. G. Mitchell.

**Investigation:** Paul Pao-Yen Wu, Toktam Babaei, Michael O'Shea.

**Methodology:** Paul Pao-Yen Wu, Kerrie Mengersen, Christopher Drovandi.

**Project administration:** Paul Pao-Yen Wu, Kerrie Mengersen, Mark A. Osborne.

**Supervision:** Paul Pao-Yen Wu, Kerrie Mengersen, Christopher Drovandi.

**Validation:** Paul Pao-Yen Wu.

**Visualization:** Paul Pao-Yen Wu, Michael O'Shea.

**Writing – original draft:** Paul Pao-Yen Wu, Toktam Babaei.

**Writing – review & editing:** Paul Pao-Yen Wu, Michael O'Shea, Kerrie Mengersen, Christopher Drovandi, Katie E. McGibbon, David B. Pyne, Lachlan J. G. Mitchell, Mark A. Osborne.

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
