## [Decision Letter · Decision Letter 0]

7 Apr 2021

PONE-D-21-03614

Predicting performance in 4 x 200-m freestyle swimming relay events

PLOS ONE

Dear Dr. Wu,

Thank you for submitting your manuscript to PLOS ONE. After careful consideration, we feel that it has merit but does not fully meet PLOS ONE’s publication criteria as it currently stands. Therefore, we invite you to submit a revised version of the manuscript that addresses the points raised during the review process.

We look forward to receiving your revised manuscript.

Kind regards,

Dalton Müller Pessôa Filho, Ph.D.

Academic Editor

PLOS ONE

Journal Requirements:

Reviewers' comments:

Reviewer's Responses to Questions

**Comments to the Author**

1. Is the manuscript technically sound, and do the data support the conclusions?

Reviewer #1: Yes

Reviewer #2: Partly

2. Has the statistical analysis been performed appropriately and rigorously? 

Reviewer #1: I Don't Know

Reviewer #2: Yes

3. Have the authors made all data underlying the findings in their manuscript fully available?

Reviewer #1: Yes

Reviewer #2: Yes

4. Is the manuscript presented in an intelligible fashion and written in standard English?

Reviewer #1: Yes

Reviewer #2: Yes

5. Review Comments to the Author

Reviewer #1: General comments

Authors are to be accomplished by selecting a very interesting topic in competitive swimming and by applying an in-depth data analysis to understand variations between individual and relay swimming performances. Manuscript is well contextualized, well written from a formal point of view and provides some interesting insights on relay performances.

Still, there are some conceptualization aspects that authors should justify and some arguments that should be further explained. Also, some of the main study limitations should be acknowledged. Please see specific comments for details.

Lastly, authors should reconsider if seven tables of results are needed to present the main results to achieve the proposed aims.

Specific comments

Introduction

Line 60-61: Is there previous research on differences between the flat start performed in individual events and the flying start performed by swimmers assigned to the second to fourth relay positions? In this case this research should be referenced. A very recent research on the topic has been published days ago (DOI: 10.1080/14763141.2021.1878262), but not many more examples of this are available…from what the present reviewer knows.

Lines 69-70: contrary to what is stated by authors, other research (http://dx.doi.org/10.1123/ijspp.2014-0577) indicates that “Highly trained swimmers do not swim (or turn) faster in relay events than in their individual races. Relay exchange times account for the difference observed in individual vs relay performance.” Maybe this should be also acknowledged on introduction.

Lines 64-76: In the present paragraph, the present reviewer considers authors are pretty “optimistic” about findings of cited studies. For example, are there enough evidence to state that relay swimmers “exert more effort than those in earlier positions”?? Authors are suggested to revise paragraph and to stick to evidence by previous studies. If additional interpretations should be inserted, authors are encouraged to used terms like “probable”, “may be”, “it is supposed”, ...

Line 99. Why 4x200m freestyle event was selected for the research purposes? This should be justified in introduction.

Methods

Line 108: what exact dates were selected for “season best time”? Natural year? September 1st to August 31th? Please specify. In the opinion of the present reviewer, one of the main weakness of the present research is that individual times could be obtained in a different season period than the major competition. Previous research has highlighted 1) the great proportion of swimmers who do not swim best times in major competitions and 2) % changes between different season periods (https://doi.org/10.1123/ijspp.2018-0782). Therefore, differences between individual and relay leg performances could be not caused by the specific relay conditions but to the different physical status of swimmers. This could be minimized by comparing the individual and relay performances within the same major competition. Race conditions would not be equal, but authors would ensure similar physical status of swimmers.

Line 112-114: The present reviewer considers it would be interesting to include exchange block times. Do these times change according to the race status or ranking for each leg? Are differences between individual and relay performances based on differences on the remaining of the race (beyond block times) or based on both block times plus the swimming laps? Considering the present research aims to “to predict and understand variations in swimmer performance between individual and relay events, and the contextual factors affecting relay team finishing positions”… does it make sense to exclude one of the main variables affecting relay races result? Authors should at least acknowledge this as a study limitation.

Line 116: what date was considered for world ranking of the swimmers in the relay? Day of major competition beginning? World ranking of the complete year? Please specify.

Line 119: 121 teams of a total of … (179 according to what indicated in line 110)?

Results

Line 213: “in the third leg the swimmers tend to swim slower than expected by 0.24 s (CI=[-0.05,0.54], p=0.10) than swimmers on the first leg”. Please rephrase.

Lines 214-216: are authors referring here to the individual or relay performance? Should “individual event” be substituted by “individual relay leg”?

Table 2: Is this table really needed in the present results section? what is the utility of the present table within the manuscript?

Table 4: Could be table 4 expressed in the text instead a table? Considering overall seven tables could distract readers from the main findings of the present research….

Discussion

The present reviewer would expect “race partial positioning” as an important variable to be included in the model to predict variations between individual and relay performances. Indeed, team tactics are usually developed according to expected partial positioning after the first, second, third leg. Are differences between individual and relay performances related to the partial positioning of relay swimmers at the beginning of their relay leg?

References

A recent reference on relay tactics (doi: 10.3389/fpsyg.2021.573285) seems to be adequate to support and discuss some of the ideas explained in the present manuscript.

Reviewer #2: General comments

The work is of interest to PLOSONE readers and a novel approach. However, some parts of the manuscript are confusing and hard to read. I recommend that you do the proposed changes and re-review it.

Specific Comments

1. Abstract. It needs to be rewritten in its entirety. Participants/sample, stasticical analysis and results are mixed. They are hard to read. Please re-write it with this in mind.

1.a) I recommend impersonal wording throughout the Abstract. Instead of "Our aim...", "The aim was..." etc. Please re-write it with this in mind.

1.b) Lines 33-35. The objective should coincide with the objective at the end of the Introduction and the beginning of the Discussion. The objective must be in the past tense as the study has already been carried out. The objective should include the term “4x200 m swimming freestyle relay events”.

1.c) Lines 35-36. “We applied linear regression and machine learning to 4 x 200-m swimming freestyle relay events”. This should be in the sentences about the statistical analysis (after participants/sample).

1.d) Line 40. “…American swimmers...” Is nationality of swimmers a studied variable? It is confusing. Information about table 1 could be included.

2. Line 82-88. It is too speculative. Please, re-write.

3. Line 98-99. Please, delete it. The objective should be the last sentence of the Introduction Section.

4. Line 135 and followings. “…a relay order of “2-1-3-4” indicates that the second fastest swimmer swam the lead-off or first leg…”. This second fastest swimmer is the second fastest according “the start time” (before the relay was swam) or “the final time” (after the relay was swam). I it can be inferred, but it needs clarification.

5. Line 155-157. Was the stepwise selection procedure used? Please, clarify

6. Please, first explain what is a “random forest” (lines 163-168) and after why was it used (lines 158-162).

7. Line 258-264. Why was the 2019 FINA World Championships used to test the model? Why not the 2012 or 2016 Olympic Games? Why was only the “battle for the 3rd place” analyzed in female? Would the results be different if other Championship was analyzed? These questions are really relevant. This information should be clarified and included in the Statistical Analysis Section.

8. The paragraphs of the Discussion section are a bit unconnected and repetitive. Please, try to make it more “readable”.

9. Line 321-324. This is a repetition of Results. Please, re-write.

10. The team position at the moment that swimmer swims could influence (very probably) in his/her time. Please, include this as a limitation.

Minor comments

11. Too much “Given…” Line 81, 84, 89… Please, re-write.

12. The Statistical Analysis Section is a bit hard to read. Please, consider to re-write it and make it more “readable”.

13. Abbreviations are used to avoid repeating words… Mean Decrease in Gini (MDG) in line 243, 252, 321…

6. PLOS authors have the option to publish the peer review history of their article (what does this mean?). If published, this will include your full peer review and any attached files.

Reviewer #1: **Yes: **Santiago Veiga

Reviewer #2: No

---

## [Author Response · Author response to Decision Letter 0]

17 May 2021

Dear Prof. Filho

Thank you to you and the reviewers for your valuable feedback. We have worked on the feedback with responses highlighted in yellow and excerpts from the updated paper highlighted in gray. Please note that the line numbers in the response refer to Manuscript.docx. Also, we have uploaded the data to supporting information.

Please see the attached response to reviewers document for the formatted response.

Thanks

Paul

PONE-D-21-03614

Predicting performance in 4 x 200-m freestyle swimming relay events

PLOS ONE

Dear Dr. Wu,

Thank you for submitting your manuscript to PLOS ONE. After careful consideration, we feel that it has merit but does not fully meet PLOS ONE’s publication criteria as it currently stands. Therefore, we invite you to submit a revised version of the manuscript that addresses the points raised during the review process.

We look forward to receiving your revised manuscript.

Kind regards,

Dalton Müller Pessôa Filho, Ph.D.

Academic Editor

PLOS ONE

 

Journal Requirements:

We have revised the format according to those guidelines.

We have uploaded the dataset used in our analysis as supporting information and updated the data availability statement. 

Upon re-submitting your revised manuscript, please upload your study’s minimal underlying data set as either Supporting Information files or to a stable, public repository and include the relevant URLs, DOIs, or accession numbers within your revised cover letter. For a list of acceptable repositories, please see http://journals.plos.org/plosone/s/data-availability#loc-recommended-repositories. Any potentially identifying patient information must be fully anonymized. Important: If there are ethical or legal restrictions to sharing your data publicly, please explain these restrictions in detail. Please see our guidelines for more information on what we consider unacceptable restrictions to publicly sharing data: http://journals.plos.org/plosone/s/data-availability#loc-unacceptable-data-access-restrictions. Note that it is not acceptable for the authors to be the sole named individuals responsible for ensuring data access.

We have added the caption to the end of the manuscript as requested

Reviewers' comments:

Reviewer's Responses to Questions

Comments to the Author

1. Is the manuscript technically sound, and do the data support the conclusions?

Reviewer #1: Yes

Reviewer #2: Partly

2. Has the statistical analysis been performed appropriately and rigorously? 

Reviewer #1: I Don't Know

Reviewer #2: Yes

3. Have the authors made all data underlying the findings in their manuscript fully available?

Reviewer #1: Yes

Reviewer #2: Yes

4. Is the manuscript presented in an intelligible fashion and written in standard English?

Reviewer #1: Yes

Reviewer #2: Yes

 

5. Review Comments to the Author

Reviewer #1: General comments

Authors are to be accomplished by selecting a very interesting topic in competitive swimming and by applying an in-depth data analysis to understand variations between individual and relay swimming performances. Manuscript is well contextualized, well written from a formal point of view and provides some interesting insights on relay performances.

Still, there are some conceptualization aspects that authors should justify and some arguments that should be further explained. Also, some of the main study limitations should be acknowledged. Please see specific comments for details.

Lastly, authors should reconsider if seven tables of results are needed to present the main results to achieve the proposed aims.

We have distilled the results and discussion now into five tables

Specific comments

Introduction

Line 60-61: Is there previous research on differences between the flat start performed in individual events and the flying start performed by swimmers assigned to the second to fourth relay positions? In this case this research should be referenced. A very recent research on the topic has been published days ago (DOI: 10.1080/14763141.2021.1878262), but not many more examples of this are available…from what the present reviewer knows.

Thank you for the reference, we have added that and this one: https://doi.org/10.1123/ijspp.2014-0577 to discuss this. 

Lines 69-70: contrary to what is stated by authors, other research (http://dx.doi.org/10.1123/ijspp.2014-0577) indicates that “Highly trained swimmers do not swim (or turn) faster in relay events than in their individual races. Relay exchange times account for the difference observed in individual vs relay performance.” Maybe this should be also acknowledged on introduction.

Thank you for the reference, we have added a discussion about it in that paragraph (line76).

However, there is conflicting evidence of differences in starts, turns and swimming speed between individual and relay events [7].

Lines 64-76: In the present paragraph, the present reviewer considers authors are pretty “optimistic” about findings of cited studies. For example, are there enough evidence to state that relay swimmers “exert more effort than those in earlier positions”?? Authors are suggested to revise paragraph and to stick to evidence by previous studies. If additional interpretations should be inserted, authors are encouraged to used terms like “probable”, “may be”, “it is supposed”, ...

Thank you for this comment – we have re-phrased that entire paragraph as suggested (line 69): 

Recently, pacing differences between individual and relay events in swimming have been examined indicating that some swimmers alter their pacing strategy during relay events [8]. This difference in pacing strategy between individual and relay swims may be attributed to the relay leg assignment as well as the added pressure to perform well for the team [8]. Extensive research on team dynamics and behavioural aspects of competitive relay swimming are described in the literature [9-12]. Compared to individual events, swimming performance is typically faster in relays which may be attributed to elevated motivation and effort [12, 13].However, there is conflicting evidence of differences in starts, turns and swimming speed between individual and relay events [7]. In addition to the motivational effects of relay swimming, the order of swimmers in the relay team can also potentially impact the effort exerted by each swimmer. Swimmers positioned in later relay leg positions were found to be more likely in certain contexts to swim faster than those in earlier positions relative to individual event times [10, 11]. Contextually, the positive influence of relay leg positioning has been ascribed to an increase in the perceived importance of individual contributions to the team outcome [11, 13]. 

Line 99. Why 4x200m freestyle event was selected for the research purposes? This should be justified in introduction.

We have clarified this on line 100:

The 4 x 200-m freestyle relay is currently the longest relay in the FINA competition schedule. With each swimmer required to complete 4 x 50 m laps the event requires well-developed speed-endurance, technical skills in starts, turns and finishes, and the element of pacing and sufficient data to model pacing effects [8]. Therefore, the aim of this study was to enhance our understanding of contextual factors contributing to relay team performance in light of individual swimmer performance, and develop predictive models to analyse the relationship between a team’s finishing position and these factors for the 4x200-m swimming freestyle relay.

Methods

Line 108: what exact dates were selected for “season best time”? Natural year? September 1st to August 31th? Please specify. In the opinion of the present reviewer, one of the main weakness of the present research is that individual times could be obtained in a different season period than the major competition. Previous research has highlighted 1) the great proportion of swimmers who do not swim best times in major competitions and 2) % changes between different season periods (https://doi.org/10.1123/ijspp.2018-0782). Therefore, differences between individual and relay leg performances could be not caused by the specific relay conditions but to the different physical status of swimmers. This could be minimized by comparing the individual and relay performances within the same major competition. Race conditions would not be equal, but authors would ensure similar physical status of swimmers.

This timeframe has been clarified to be the same season, typically beginning around September and ending around July/August (line 116). 

For each relay swimmer, the individual 200-m freestyle season's best time for the same season (typically beginning around September and concluding around July-August) was located using FINA world rankings (https://www.fina.org/content/swimming-world-ranking).

Although the use of the best time within the competition could improve predictions, that would the limit the ability to use the model to assist in team selection, training and preparation, as those activities occur weeks to months prior to the competition (Introduction line 86). In addition, each country is typically only allowed two competitors per individual event at major competitions such as the Olympics and World Championships, thus the best swim time in the same competition would only be available for half of the relay team. 

(line 341) The ability to accurately predict team finishing position based on a set of explanatory variables would support coaches in making an evidence-based decision when selecting relay team swimmers and leg assignments weeks to months ahead of competition. 

Season best time as currently construed represents more so the calibre of the swimmer rather than the current physical status; however, understanding physical status is a key area for future work. Both points have been clarified in the discussion. 

(line 387) However, increasingly more data are becoming available as demonstrated by the availability of each swimmer’s season’s best time and world ranking going into the 2019 World Championships. Potentially, data about individual swimmer’s physical status or performance characteristics (such as individual and relay block times [19]) could be used to extend this work and improve the predictivity of the model.

Line 112-114: The present reviewer considers it would be interesting to include exchange block times. Do these times change according to the race status or ranking for each leg? Are differences between individual and relay performances based on differences on the remaining of the race (beyond block times) or based on both block times plus the swimming laps? Considering the present research aims to “to predict and understand variations in swimmer performance between individual and relay events, and the contextual factors affecting relay team finishing positions”… does it make sense to exclude one of the main variables affecting relay races result? Authors should at least acknowledge this as a study limitation.

In our study, differences between individual and relay performances are effectively based on swimming time only (i.e. without block times) (Methods 121). 

We agree that investigating how relay exchange block times might change with race status, or ranking, might be interesting and we have reflected that in the discussion for future investigation. However, such a model should capture the inherent dependency of block times on individual swimmers, which requires multiple measurements per individual. This is challenging with the current dataset as the median number of relay races per swimmer is 1, and less than 25% of swimmers in this dataset had 3 or more recorded relay races. More data is needed to be able to analyse block times in a statistically rigorous manner. This point has been clarified in the Discussion section (line 389).

Data about individual swimmer’s physical status or performance characteristics (such as individual and relay block times [19]) could be used to extend this work and improve the predictivity of the model. Currently, less than 25% of the swimmers in this dataset had 3 or more relay races recorded and the majority only had 1; this limited our ability to model individual characteristics.

Line 116: what date was considered for world ranking of the swimmers in the relay? Day of major competition beginning? World ranking of the complete year? Please specify.

The world ranking has been clarified as follows (line 125):

The team average ranking was calculated as the average world ranking of the four swimmers in the team, where world rankings for the year of the relay competition were used. In our dataset, the swims contributing to the rankings were coincidentally prior to the major competition of that year. 

Line 119: 121 teams of a total of … (179 according to what indicated in line 110)?

121 teams of a total of 188. (line 131)

Results

Line 213: “in the third leg the swimmers tend to swim slower than expected by 0.24 s (CI=[-0.05,0.54], p=0.10) than swimmers on the first leg”. Please rephrase.

(line 234) In addition, compared to the first leg, swimmers in the third leg tend to swim slower than expected by 0.24 s (CI=[-0.05,0.54], p=0.10).

Lines 214-216: are authors referring here to the individual or relay performance? Should “individual event” be substituted by “individual relay leg”?

We are referring to the individual 200m freestyle event (clarified, line 236)

Table 2: Is this table really needed in the present results section? what is the utility of the present table within the manuscript?

We agree that Table 2 is not essential and have removed it. 

Table 4: Could be table 4 expressed in the text instead a table? Considering overall seven tables could distract readers from the main findings of the present research….

We have removed Table 4 and placed this as text in the Results (line 259).

Discussion

The present reviewer would expect “race partial positioning” as an important variable to be included in the model to predict variations between individual and relay performances. Indeed, team tactics are usually developed according to expected partial positioning after the first, second, third leg. Are differences between individual and relay performances related to the partial positioning of relay swimmers at the beginning of their relay leg?

We agree that race partial positioning is an important variable; however, it presents an additional layer of modelling complexity, predicting three intermediary and one final position, which typically requires more data for fitting and validation. 

This point has been added to the Discussion line 399:

Additionally, intermediary positions and whether the team is still in medal contention or not at the end of each leg could also affect individual swimmer performance. Multivariate extensions of the proposed models could be developed, with sufficient data, for this more complex prediction task.

References

A recent reference on relay tactics (doi: 10.3389/fpsyg.2021.573285) seems to be adequate to support and discuss some of the ideas explained in the present manuscript.

Thank you for this recommendation. We have incorporated this recent work into the manuscript in the following areas: 

(line 58) A key challenge of relay events in sporting competitions is team selection and the order of athletes, as they can impact race outcomes [1-3].

(line 66) However, in both track running and swimming, it appears that selecting the fastest athlete for the first or lead-off relay leg is popular and successful [1-3], although further research is required to determine how this impacts team performance.

(line 402) Finally, hybrids of machine learning and statistical methods could be used in future research to build similar predictive models for other swimming relay events, such as the mixed relays [3]. 

Reviewer #2: General comments

The work is of interest to PLOSONE readers and a novel approach. However, some parts of the manuscript are confusing and hard to read. I recommend that you do the proposed changes and re-review it.

Specific Comments

1. Abstract. It needs to be rewritten in its entirety. Participants/sample, stasticical analysis and results are mixed. They are hard to read. Please re-write it with this in mind.

1.a) I recommend impersonal wording throughout the Abstract. Instead of "Our aim...", "The aim was..." etc. Please re-write it with this in mind.

We took the opportunity to review carefully the content and wording of the abstract with reference to the reviewer’s comment – our abstract has the following structure: 

Aim (sentence 1)

Method (sentence 2)

Data (sentence 3, 4)

Results (sentence 5, 6)

Discussion (sentence 7-10)

We have added headings in the abstract to help separate out the sections, and made the abstract impersonal as suggested:

Abstract

Aim

The aim was to predict and understand variations in swimmer performance between individual and relay events, and develop a predictive model for the 4x200-m swimming freestyle relay event to help inform team selection and strategy. 

Data and Methods

Race data for 716 relay finals (4 x 200-m freestyle) from 14 international competitions between 2010-2018 were analysed. Individual 200-m freestyle season best time for the same year was located for each swimmer. Linear regression and machine learning was applied to 4 x 200-m swimming freestyle relay events.

Results

Compared to the individual event, the lowest ranked swimmer in the team (-0.62 s, CI=[-0.94,-0.30]) and American swimmers (-0.48 s [-0.89,-0.08]) typically swam faster 200-m times in relay events. Random forest models predicted gold, silver, bronze and non-medal with 100%, up to 41%, up to 63%, and 93% sensitivity, respectively. 

Discussion

Team finishing position was strongly associated with the differential time to the fastest team (mean decrease in Gini (MDG) when this variable was omitted =31.3), world rankings of team members (average ranking MDG of 18.9), and the order of swimmers (MDG=6.9). Differential times are based on the sum of individual swimmer’s season’s best times, and along with world rankings, reflect team strength. In contrast, the order of swimmers reflects strategy. This type of analysis could assist coaches and support staff in selecting swimmers and team orders for relay events to enhance the likelihood of success.

1.b) Lines 33-35. The objective should coincide with the objective at the end of the Introduction and the beginning of the Discussion. The objective must be in the past tense as the study has already been carried out. The objective should include the term “4x200 m swimming freestyle relay events”.

We have ensured consistency of wording across the Abstract, Introduction and Discussion:

Abstract (line 34)

The aim was to predict and understand variations in swimmer performance between individual and relay events, and develop a predictive model for the 4x200-m swimming freestyle relay event to help inform team selection and strategy.

Introduction (line 104)

Therefore, the aim of this study was to enhance our understanding of contextual factors contributing to relay team performance in light of individual swimmer performance, and develop predictive models to analyse the relationship between a team’s finishing position and these factors for the 4x200-m swimming freestyle relay.

Discussion (line 291)

The statistical approaches developed in this study were useful in identifying the variables affecting relay swimming performance given individual swimmer performance, and predicting relay team finishing positions for the 4x200-m freestyle relay.

1.c) Lines 35-36. “We applied linear regression and machine learning to 4 x 200-m swimming freestyle relay events”. This should be in the sentences about the statistical analysis (after participants/sample).

Data (participants/sample) has been placed before methods in the abstract as suggested.

1.d) Line 40. “…American swimmers...” Is nationality of swimmers a studied variable? It is confusing. Information about table 1 could be included.

Yes, in addition to the abstract, nationality is listed as a variable in Table 1, included as a variable in equations 2 and 3, with nationality effects presented in the Results and further analysed in the Discussion.

2. Line 82-88. It is too speculative. Please, re-write.

We have clarified this paragraph by removing the sentence beginning with “Given this long period between selection and the major competition…”, and added a citation for the statement about physiological, psychological and team-based dynamics (line 84):

During FINA-sanctioned events including the biennial World Championships, relay teams must nominate their four selected swimmers, and the team order, one hour prior to the start of the heats or finals session in which the relay occurs [15]. However, swimmers are typically selected for the national squad a few weeks to several months prior to the competition based on their performance in the corresponding individual event. In addition, due to the complex interactions between physiological, psychological and team-based dynamics [10], predictions of individual performance in relays and overall team outcomes are challenging. Therefore, there is a need for effective predictive tools that could support coaches in the decision-making process to maximise the performance of the relay team as a whole, as well as each individual swimmer. 

3. Line 98-99. Please, delete it. The objective should be the last sentence of the Introduction Section.

Deleted.

4. Line 135 and followings. “…a relay order of “2-1-3-4” indicates that the second fastest swimmer swam the lead-off or first leg…”. This second fastest swimmer is the second fastest according “the start time” (before the relay was swam) or “the final time” (after the relay was swam). I it can be inferred, but it needs clarification.

Here, we are using the swimmer’s world ranking and have clarified this point in line 148: 

The order of swimmers in the relay was encoded according to the relative world ranking of each swimmer within a team. For example, a relay order of “2-1-3-4” indicates that the second fastest swimmer swam (i.e. second highest world ranking) the lead-off or first leg, the fastest swimmer swam the second leg, and so on.

5. Line 155-157. Was the stepwise selection procedure used? Please, clarify

No it was not, and this point has been clarified on line 166:

Multiple linear regression [21] was used to estimate the relationships between an individual swimmer’s performance in a relay and the explanatory variables (Eq 2).

6. Please, first explain what is a “random forest” (lines 163-168) and after why was it used (lines 158-162).

We have re-ordered the text as suggested (line 170):

Random forests were used to predict team finishing positions based on explanatory variables as they are ideally suited for a mixture of numeric and categorical variables with potentially highly non-linear relationships. Random forests are an ensemble modelling extension of simple decision trees, which recursively partition the space of explanatory variables to minimise some dispersion criteria (i.e. measure of variability) in the resultant partitions [22]. Random forests have also demonstrated high predictive sensitivity and specificity for complex problems in many domains [22]. This method helps to overcome the overfitting problem encountered in decision trees by building many shallow trees using data subsets sampled through bagging. We built a random forest model, referred to as RF1, to predict gold, silver, bronze or non-medal finishing positions. To assist with better prediction of medal colour, we also trialled a model that only predicts medal colour, RF2.

7. Line 258-264. Why was the 2019 FINA World Championships used to test the model? Why not the 2012 or 2016 Olympic Games? Why was only the “battle for the 3rd place” analyzed in female? Would the results be different if other Championship was analyzed? These questions are really relevant. This information should be clarified and included in the Statistical Analysis Section.

There has been a misunderstanding. The random forest model was tested using leave-one-out cross-validation on all of the competitions available in the dataset, including the 2012 and 2016 Olympic Games. Leave-one-out cross-validation ensures that we train the model on all other competitions, and test on a competition that was not used for training the model. The aggregated result was a model (RF1) that was highly effective at correctly predicting gold medal winning teams (100% sensitivity) and whether a team will medal or not (non-medalling sensitivity of 93%) (line 295). For further details, please refer to Statistical Analysis line 200, Results line 257, Table 3, and Discussion line 295. 

However, to help illustrate how the model could be used, a case study was performed on particular aspects of the 2019 FINA World Championships as they were the most recent competition. We have clarified these points in the Statistical Analysis (line 215) 

Finally, the utility of the model was demonstrated by applying it to a case study analysis of the 2019 World Championships. 

and the Discussion (line 358):

To illustrate how the model could be used to support decision making, we demonstrate with a case study of predicting the finishing positions for the top four teams at the 2019 FINA World Championships. 

8. The paragraphs of the Discussion section are a bit unconnected and repetitive. Please, try to make it more “readable”.

We took the opportunity to revisit the sequence of paragraphs in the Discussion section, removing repetitive statements and clarifying throughout. There are 6 sub-sections within the Discussion, which have been explicitly labelled with sub-headings as follows: 1) opening paragraph summarising main outcomes and applications, 2) differentiating psychological from technical effects, 3) nationality issues, primarily USA, 4) relative influence of variables, especially the issue of swimmer ranking with cross-reference to the literature and individual performances, 5) an illustrative case study of how the model could be applied, 6) limitations and future work.

Discussion

The statistical approaches developed in this study were useful in identifying the variables affecting relay swimming performance given individual swimmer performance, and predicting relay team finishing positions for the 4x200-m freestyle relay. Results indicate that swimmers from the USA, and those swimmers who were the slowest within their teams according to ranking, typically performed better in relays than in individual events. The random forest model RF1 was highly effective at correctly predicting gold medal winning teams (100% sensitivity), and whether a team will medal or not (non-medalling sensitivity of 93%). However, the models were less accurate in distinguishing between silver (35% using RF1, 41% using RF2) and bronze (13% using RF1, 63% using RF2). This outcome might be due to small differential times between these positions for some swimming competitions. In contrast, the differential times between the bronze medal position and non-medal positions for all competitions tended to be much larger. The RF2 model could be used by decision makers to evaluate silver and bronze medal scenarios assuming that a team will win a medal. These models enable coaches and support staff to simulate different relay race scenarios to determine the optimal relay team configuration by using swimmer characteristics, anticipated opponent swimmers and team order. 

Differentiating Psychological from Technical Effects

Among the many variables that may impact relay swimming performance, the psychology of team competition is important [13, 14]. Note that we have adjusted for the effect of the flying start in relay legs two through four by setting exchange block times equal to individual reaction time [8]. Any residual differences between legs were captured via the relay leg term; thus, we were able to discern potential psychological effects from technical effects. 

Our results indicate that the largest effect of the variables modelled in this study was due to the worst-ranked or slowest swimmer in a team. These swimmers typically swam 0.62 s faster in the relay than in the corresponding individual event. Peer effects can have a positive impact on individual performance within a team, and these psychosocial effects may help explain the improved performance of some swimmers in relays relative to their individual times in the present study [10, 25]. However, our findings differ from those of Hüffmeier and Hertel (12) who reported on the effects of relay leg assignment (i.e. going first, second, third or last). In contrast, we found the relative ranking of the swimmer within the team (i.e. worst-ranked swimmer) to be a larger effect, and relay leg assignment to be generally not significant. Motivating group effects are typically greater when an individual perceives their contribution as important to the overall team outcome [12, 14]. Therefore, it is possible that the slower swimmers within the team felt more pressure and motivation to step up and put their team in a good position. In contrast, relay teams comprised of higher ranking athletes are more likely to underperform relative to their individual performance [25]. Such psychological impacts could be an area for further study to help motivate and develop swimmers in relay and non-relay contexts. 

Nationality Impacts

Swimmer nationality also impacted performance as individual swimmers from the USA tended to swim 0.48 s faster during the relays than their predicted individual swim times. This outcome could be attributed to the competition structure of the National Collegiate Athletic Association (NCAA) which allows for the frequent practise of relay swimming in competitive races. In contrast, Australian swimmers (and those of many other nations) may only swim in a limited number of relay events throughout the season prior to the major international competition, and rarely get the opportunity to practice with potential teammates. Team cohesiveness may play a role as social loafing is less likely to occur in highly cohesive teams [26]. However, further research is required to determine the underlying nature of differences between nations. 

Relative Influence of Variables

The ability to accurately predict team finishing position based on a set of explanatory variables would support coaches in making an evidence-based decision when selecting relay team swimmers and leg assignments, potentially weeks to months ahead of competition. Random forest models were used to make these predictions and the most influential variables were identified based on cross-validation, and the mean decrease in sensitivity and specificity as measured by MDG [22]. As might be expected, the strength of the team, as captured by rankings and individual season’s best times, was the leading contributor to finishing position (Results). However, team strategy, in terms of the order of swimmers was the next most influential factor. The dataset used for modelling comprised primarily of high calibre, international events including Olympics and World Championships. Typically, these are the pinnacle events that athletes train and prepare for. We identified that medal outcomes were highly influenced by differential time (MDG of 31.3), which is based on the sum of individual swimmer’s season’s best times. This outcome suggests that individuals are performing at or near their best at these international relay competitions and, equivalently, that season’s best times are useful in predicting individual swimmers’ performance at pinnacle relay events. 

Illustrative Case Study

To illustrate how the model could be used to support decision making, we demonstrate with a case study of predicting the finishing positions for the top four teams at the 2019 FINA World Championships. This data, which included world rankings and season best times coming into the competition, were not included in the original dataset. Although the gold medal predictions were correct, the model incorrectly predicted the bronze and 4th positions for females, and silver and bronze positions for males. Team average ranking for the two female teams was identical with a similar differential between the highest and lowest ranked swimmer. However, the fourth placed team had the best ranking swimmer overall, which may indicate that this team underperformed relative to their expected team performance time. This explanation may also serve as a reason for the incorrect model prediction here for both medal colour and medal or non-medal. Similarly, the USA Men’s team was predicted to finish in the silver medal position, but Russia outperformed them by just 0.17 s. However, the model was able to correctly predict a medal and non-medal position.

These models can also be used in a predictive decision support scenario where the impact of different swimmer orders on finishing position can be evaluated in a risk-informed, probabilistic manner. For the Canadian women’s teams in the 2019 FINA World Championships, the order used in the race provided the highest chance of bronze and lowest chance of a non-medal finish. A 2xx1 order could have increased the chance for a silver medal by 9.4%, but also increase the chance for missing out on a medal by 12.5%. According to the model, China would have increased their chance of a bronze medal and slightly decreased their chance of a non-medal finish if they applied another swimmer order or 21xx. However, these scenarios only serve as illustrations, and should be seen as observations given limitations of the data, the numerous possible swimmer order and ranking combinations, and the many other factors influencing medal finishes that were not included in the model.

Limitations and Future Work

While these statistical approaches were successfully applied to enhance our understanding of the variables impacting both individual and team performance in relay swimming events, there are some limitations. First, only teams with available data for all four swimmers were analysed, resulting in partial data for some races which is a potential source of misclassification errors. However, increasingly, more data are becoming available as demonstrated by the availability of each swimmer’s season’s best time and world ranking going into the 2019 World Championships. Potentially, data about individual swimmer’s physical status or performance characteristics (such as individual and relay block times [20]) could be used to extend this work and improve the predictivity of the model. Currently, less than 25% of the swimmers in this dataset had 3 or more relay races recorded and the majority only had 1; this limited our ability to model individual characteristics. 

We also assumed that all relay teams had an equal chance of finishing in each position, whereas in reality some teams are more likely to be chasing medal positions than others. Incorporating such prior knowledge into the model, such as through a Bayesian formulation, could strengthen the prediction accuracy. Moreover, the individual analyses could be combined through a hierarchical model to enable ‘borrowing of strength’ between events to improve estimates and extend insights. Additionally, intermediary positions and whether the team is still in medal contention or not at the end of each leg can also potentially affect individual swimmer performance. Multivariate extensions of the proposed models could potentially be developed, with sufficient data, for this, more complex, task. Finally, hybrids of machine learning and statistical methods could be used in future research to build similar predictive models for other swimming relay events, such as mixed relay events [3]. 

9. Line 321-324. This is a repetition of Results. Please, re-write.

We focused that on the importance of strength of team, followed by order of swimmers (line 346):

As might be expected, the strength of the team, as captured by rankings and individual season’s best times, was the leading contributor to finishing position (Results). However, team strategy, in terms of the order of swimmers was the next most influential factor.

10. The team position at the moment that swimmer swims could influence (very probably) in his/her time. Please, include this as a limitation.

This point has been added to the discussion line 399:

Additionally, intermediary positions and whether the team is still in medal contention or not at the end of each leg can also potentially affect individual swimmer performance. Multivariate extensions of the proposed models could potentially be developed, with sufficient data, for this, more complex, prediction task.

Minor comments

11. Too much “Given…” Line 81, 84, 89… Please, re-write.

We have replaced the repetitions with “due to the…” and “with an…” (line 88 and line 94).

12. The Statistical Analysis Section is a bit hard to read. Please, consider to re-write it and make it more “readable”.

This section has been clarified and re-structured with subheadings and an opening text outlining the structure has been provided:

Statistical Analysis 

Two main types of methods were used: (i) linear regression, to study individual swimmer’s relay performances, and (ii) random forests, to predict race outcomes given team configurations. This section describes the two methods, model fitting and model validation. 

Linear Regression

Multiple linear regression [21] was used to estimate the relationships between an individual swimmer’s performance in a relay and the explanatory variables (Eq 2). An explanatory variable was deemed to have a significant effect if p≤0.05.

Random Forests

Random forests were used to predict team finishing positions based on explanatory variables as they are ideally suited for a mixture of numeric and categorical variables with potentially highly non-linear relationships. Random forests are an ensemble modelling extension of simple decision trees, which recursively partition the space of explanatory variables to minimise some dispersion criteria (i.e. measure of variability) in the resultant partitions [22]. Random forests have also demonstrated high predictive sensitivity and specificity for complex problems in many domains [22]. This method helps to overcome the overfitting problem encountered in decision trees by building many shallow trees using data subsets sampled through bagging. We built a random forest model, referred to as RF1, to predict gold, silver, bronze or non-medal finishing positions. To assist with better prediction of medal colour, we also trialled a model that only predicts medal colour, RF2. We developed a predictor variable based on the observation that team performance in a relay is the sum of the individual performance times of the four swimmers within the team. Therefore, based on the sum of the season’s best individual times we constructed a theoretical performance measure of each team relative to the theoretical performance of the fastest team based on differential time (Diff.Time) defined as follows:

 Diff.Time_j=∑_(i=1)^4▒s_ij -min┬(∀j)⁡∑_(i=1)^4▒s_ij (1)

where for team j and individual i, s_ij is the season’s best time for that swimmer. 

Model Fitting

All statistics were calculated using R software [23] and implemented with the base and randomForest packages to fit linear regression and random forest models, respectively. The parameters of the random forest were tuned by making use of a cross-validation based technique. Five-fold cross validation was run 100 times in conjunction with a grid search for selecting model parameters including the number of variables to sample at each split in the tree, and the number of variables sampled as candidates at each split in the tree. Given the randomly sampled nature of random forests, repeated evaluations provide a more robust selection for the tuning parameters [24].

Model Validation

For the linear regression model, goodness of fit is sufficient to give confidence that the model is reasonable, and the model can be interrogated to ascertain the impact of different explanatory variables on individual performance [21]. In comparison, the random forest was employed to predict race finishing position, so we validated model performance using leave-one-out cross-validation. In this scheme, we iterated over each data point, trained with all other data points and tested with the current data point. 

We used a 4x4 confusion matrix to show the number of times a recorded gold, silver, bronze or non-medal result (corresponding to rows) was classified by the model as a gold, silver, bronze or non-medal outcome (columns corresponded to predictions). We computed model sensitivity, also referred to as producer’s accuracy when there are more than two categories, which is the rate at which the model correctly classifies a result as a member of a certain category [24]. Note that there is no direct analogue for specificity when there are more than two categories. The randomForest package uses the Gini index as one approach to capture both sensitivity and specificity [22]. This index is useful for assessing both the validity of the model, and for quantifying the relative influence of explanatory variables based on the decrease in the Gini index when a variable is removed from the model. 

Finally, the utility of the random forest was demonstrated by applying it to a case study analysis of the 2019 World Championships. 

13. Abbreviations are used to avoid repeating words… Mean Decrease in Gini (MDG) in line 243, 252, 321…

We have corrected the text to use MDG

6. PLOS authors have the option to publish the peer review history of their article (what does this mean?). If published, this will include your full peer review and any attached files.

Do you want your identity to be public for this peer review? For information about this choice, including consent withdrawal, please see our Privacy Policy.

Reviewer #1: Yes: Santiago Veiga

Reviewer #2: No

---

## [Decision Letter · Decision Letter 1]

15 Jun 2021

PONE-D-21-03614R1

Predicting performance in 4 x 200-m freestyle swimming relay events

PLOS ONE

Dear Dr. Wu,

Thank you for submitting your manuscript to PLOS ONE. After careful consideration, we feel that it has merit but does not fully meet PLOS ONE’s publication criteria as it currently stands. Therefore, we invite you to submit a revised version of the manuscript that addresses the points raised during the review process.

We look forward to receiving your revised manuscript.

Kind regards,

Dalton Müller Pessôa Filho, Ph.D.

Academic Editor

PLOS ONE

Journal Requirements:

Additional Editor Comments (if provided):

The reviewers congratulate the authors on the improvement of this manuscript after the first round of revision. However, Reviewer #2 has addressed two new comments to the authors. Therefore, the authors need to provide the responses to these other comments before the final decision.

Reviewers' comments:

Reviewer's Responses to Questions

**Comments to the Author**

1. If the authors have adequately addressed your comments raised in a previous round of review and you feel that this manuscript is now acceptable for publication, you may indicate that here to bypass the “Comments to the Author” section, enter your conflict of interest statement in the “Confidential to Editor” section, and submit your "Accept" recommendation.

Reviewer #1: All comments have been addressed

Reviewer #2: All comments have been addressed

2. Is the manuscript technically sound, and do the data support the conclusions?

Reviewer #1: (No Response)

Reviewer #2: Yes

3. Has the statistical analysis been performed appropriately and rigorously? 

Reviewer #1: (No Response)

Reviewer #2: Yes

4. Have the authors made all data underlying the findings in their manuscript fully available?

Reviewer #1: (No Response)

Reviewer #2: Yes

5. Is the manuscript presented in an intelligible fashion and written in standard English?

Reviewer #1: (No Response)

Reviewer #2: Yes

6. Review Comments to the Author

Reviewer #1: Authors have adequately addressed the comments raised in a previous round of review. However, it was difficult for the present reviewer to find the authors’ responses within the text. Authors should indicate their responses by bullet points or “response”.

Reviewer #2: General comments

Thank you for accepting the suggestions. I would like to suggest two more things:

The objective should coincide with the Abstract (lines 33-36) and the Introduction (lines 104-107). To predict and to enhance are not the same.

Line 101-104. Please, clarify more why the study was done in 4 x 200-m swimming freestyle relay events

7. PLOS authors have the option to publish the peer review history of their article (what does this mean?). If published, this will include your full peer review and any attached files.

Reviewer #1: **Yes: **Santi Veiga

Reviewer #2: No

---

## [Author Response · Author response to Decision Letter 1]

24 Jun 2021

Dear Prof. Filho

Thank you to you and the reviewers for your feedback. Please see below our responses and attached our formatted response to reviewers, where responses are highlighted in yellow and excerpts from the updated paper highlighted in gray. Please note that the line numbers in the response refer to Manuscript.docx. 

Thanks

Authors

Comments to the Author

1. If the authors have adequately addressed your comments raised in a previous round of review and you feel that this manuscript is now acceptable for publication, you may indicate that here to bypass the “Comments to the Author” section, enter your conflict of interest statement in the “Confidential to Editor” section, and submit your "Accept" recommendation.

Reviewer #1: All comments have been addressed

Reviewer #2: All comments have been addressed

2. Is the manuscript technically sound, and do the data support the conclusions?

Reviewer #1: (No Response)

Reviewer #2: Yes

3. Has the statistical analysis been performed appropriately and rigorously? 

Reviewer #1: (No Response)

Reviewer #2: Yes

4. Have the authors made all data underlying the findings in their manuscript fully available?

Reviewer #1: (No Response)

Reviewer #2: Yes

5. Is the manuscript presented in an intelligible fashion and written in standard English?

Reviewer #1: (No Response)

Reviewer #2: Yes

6. Review Comments to the Author

Reviewer #1: Authors have adequately addressed the comments raised in a previous round of review. However, it was difficult for the present reviewer to find the authors’ responses within the text. Authors should indicate their responses by bullet points or “response”.

Reviewer #2: General comments

Thank you for accepting the suggestions. I would like to suggest two more things:

The objective should coincide with the Abstract (lines 33-36) and the Introduction (lines 104-107). To predict and to enhance are not the same.

We have made the Introduction consistent with the Abstract (line 105):

Therefore, the aim of this study was to predict and better understand contextual factors contributing to relay team performance in light of individual swimmer performance, and develop predictive models to analyse the relationship between a team’s finishing position and these factors for the 4x200-m swimming freestyle relay.

Line 101-104. Please, clarify more why the study was done in 4 x 200-m swimming freestyle relay events

We have clarified as follows (line 101):

With each swimmer required to complete 4 x 50 m laps the event requires well-developed speed-endurance, technical skills in starts, turns and finishes, and the element of pacing and sufficient data to model pacing effects [9]. This complexity makes the 4 x 200-m freestyle ideal as a starting point for developing and testing predictive and analytical tools.

---

## [Decision Letter · Decision Letter 2]

29 Jun 2021

Predicting performance in 4 x 200-m freestyle swimming relay events

PONE-D-21-03614R2

Dear Dr. Wu,

We’re pleased to inform you that your manuscript has been judged scientifically suitable for publication and will be formally accepted for publication once it meets all outstanding technical requirements.

Kind regards,

Dalton Müller Pessôa Filho, Ph.D.

Academic Editor

PLOS ONE

Additional Editor Comments (optional):

Reviewer #2 has accepted the manuscript for publication in Plos One. Congratulations to the authors!

Reviewers' comments:

Reviewer's Responses to Questions

**Comments to the Author**

1. If the authors have adequately addressed your comments raised in a previous round of review and you feel that this manuscript is now acceptable for publication, you may indicate that here to bypass the “Comments to the Author” section, enter your conflict of interest statement in the “Confidential to Editor” section, and submit your "Accept" recommendation.

Reviewer #2: All comments have been addressed

2. Is the manuscript technically sound, and do the data support the conclusions?

Reviewer #2: Yes

3. Has the statistical analysis been performed appropriately and rigorously? 

Reviewer #2: Yes

4. Have the authors made all data underlying the findings in their manuscript fully available?

Reviewer #2: Yes

5. Is the manuscript presented in an intelligible fashion and written in standard English?

Reviewer #2: Yes

6. Review Comments to the Author

Reviewer #2: (No Response)

7. PLOS authors have the option to publish the peer review history of their article (what does this mean?). If published, this will include your full peer review and any attached files.

Reviewer #2: No

---

## [Editor Report · Acceptance letter]

2 Jul 2021

PONE-D-21-03614R2 

Predicting performance in 4 x 200-m freestyle swimming relay events 

Dear Dr. Wu:

I'm pleased to inform you that your manuscript has been deemed suitable for publication in PLOS ONE. Congratulations! Your manuscript is now with our production department. 

Kind regards, 

on behalf of

Prof. Dr. Dalton Müller Pessôa Filho 

Academic Editor

PLOS ONE